# Entropy Perspectives of Molecular and Evolutionary Biology

**DOI:** 10.3390/ijms23084098

**Published:** 2022-04-07

**Authors:** Bartolomé Sabater

**Affiliations:** Department of Life Sciences, University of Alcalá, 28805 Alcalá de Henares, Madrid, Spain; bartolome.sabater@uah.es; Tel.: +34-609-227-010

**Keywords:** cancer, DNA informational entropy, cell compartmentation, evolutionary biology, lactate dehydrogenase (LDH), lactic acid, metabolism, thermodynamic entropy, Warburg effect

## Abstract

Attempts to find and quantify the supposed low entropy of organisms and its preservation are revised. The absolute entropy of the mixed components of non-living biomass (approximately −1.6 × 10^3^ J K^−1^ L^−1^) is the reference to which other entropy decreases would be ascribed to life. The compartmentation of metabolites and the departure from the equilibrium of metabolic reactions account for reductions in entropy of 1 and 40–50 J K^−1^ L^−1^, respectively, and, though small, are distinctive features of living tissues. DNA and proteins do not supply significant decreases in thermodynamic entropy, but their low informational entropy is relevant for life and its evolution. No other living feature contributes significantly to the low entropy associated with life. The photosynthetic conversion of radiant energy to biomass energy accounts for most entropy (2.8 × 10^5^ J K^−1^ carbon kg^−1^) produced by living beings. The comparatively very low entropy produced in other processes (approximately 4.8 × 10^2^ J K^−1^ L^−1^ day^−1^ in the human body) must be rapidly exported outside as heat to preserve low entropy decreases due to compartmentation and non-equilibrium metabolism. Enzymes and genes are described, whose control minimizes the rate of production of entropy and could explain selective pressures in biological evolution and the rapid proliferation of cancer cells.

## 1. Introduction

Thermodynamically, organisms are open systems that maintain their assumed low entropy by exporting the metabolically produced entropy as heat [1,2]. The production, influx, and outflux rates of entropy by the whole organism have been frequently determined experimentally and estimated theoretically (see [3] for a review). However, there are uncertainties about the magnitude of entropy content of organisms, the value to which low entropy is compared, the molecules to which low entropy is associated, and about the relative contribution of each living reaction to generate or save entropy.

Bioenergetic investigations have been mainly focused on values of and changes in Gibbs free energy (*G*, Δ*G*) and enthalpy (*H*, Δ*H*), which have a more evident physiological significance than entropy (*S*, Δ*S*), which, in this regard, may be approached as indicative of how much enthalpy cannot be recovered as free energy according to the relations:*G* = *H* − *T* × *S* and Δ*G* = Δ*H* − *T* × Δ*S*

for actual values and changes (Δ), respectively, in processes at the absolute temperature *T*.

Free energy and enthalpy have a clear physical significance, which determines the course of biological processes related to equilibrium constants of reactions and energy requirements [4]. However, a role of entropy per se has barely been assigned in biology. Recent theoretical and experimental investigations are uncovering aspects of development, cancer, and biological evolution, the understanding of which benefits from entropy approaches and, furthermore, entropy content and changes determine their occurrence.

For well-defined chemical components, the absolute entropy of formation from their constituting atomic elements [4] (https://homepages.wmich.edu, accessed on 13 March 2022) has become the most common reference. However, the low entropy associated with one organism structure has been approached diversely. Sometimes, the low entropy of one macro-structure is related to the entropy of their disassembled molecules. Frequently, this refers to the magnitude of entropy produced when the components of the organism are oxidized to CO_2_ + H_2_O. Usually, for well-defined chemicals and for reactions, the entropy content and production (respectively) are expressed per mole. However, for some specific purposes, they are expressed per carbon atom gram, total mass, volume and, even, energy content or produced involved.

When referring to one unit of carbon weight, the entropy of formation at room temperature (25–30 °C) of the dry matter of cells is in the same range as that of glucose and amino acids commonly feeding the growth of cells [3,5], approximately 2 × 10^3^ J K^−1^ carbon kg^−1^, and of the same living cell [6], and lower than CO_2_ gas (4.8 × 10^3^ J K^−1^ carbon kg^−1^). Therefore, in addition to chemical biomass, some living features should account for minor contributions to the low entropy of organisms. The compartmentation of components, sequences of nucleic acids and proteins, ordered membrane structures, etc., although key to life, has minor contributions to the low relative entropy of the whole organism. In fact, the standard entropy of formation of dry glucose, −2.12 × 10^2^ J K^−1^ mol^−1^ [7] (equivalent to −2.95 × 10^3^ J K^−1^ carbon kg^−1^), decreases in aqueous solution to −1.16 × 10^3^ J K^−1^ glucose mol^−1^ (calculated from [4]), equivalent to −6.44 × 10^3^ J K^−1^ glucose kg^−1^ or −1.61 × 10^4^ J K^−1^ carbon kg^−1^. Considering that carbon accounts for approximately 9% (*w*/*w*) of fresh living matter (one liter, L, weighting 1.1 kg), this has −1.6 × 10^3^ J K^−1^ L^−1^ (−1.61 × 10^4^ × 1.1 × 9/100) entropy attributable to biomass standard formation in situ.

The comparison of the entropy of chemical components and structures with the rate of entropy production in different metabolic reactions and with the entropy fluxes in organisms permits an approximate evaluation of the role of entropy export, and the contribution of structure/function entropies to support life and biological evolution.

## 2. Entropy Fluxes in Photosynthesis

Plants, where radiant energy plays a key role in energy and entropy fluxes, are a useful model to compare a wide range of entropy fluxes with low-entropy reservoirs and physiological processes. Yourgrau and Van Der Merwe [8] clearly demonstrated that plant photosynthesis corroborates the thermodynamics second principle of the increase in entropy and, despite more recent polemics related to the primary photochemical stages [9,10], the increase in entropy is widely accepted for the entire process and all stages of photosynthesis [11,12].

Starting from the low-entropy energy of the absorbed light, its full conversion to the high-entropy of heat energy is diminished by successive stages of the use, storage, and export of energy by plants that, by synthesizing low-entropy chemicals, decrease the export of entropy as heat. Potential entropy is trapped in radiant energy, and photosynthesis captures part of this potential entropy (sometimes named negentropy, [13]) tied to the free energy of the biosynthesized chemicals. Energy, as heat, and associated entropy are released from these chemicals through respiratory firing in the same plants or in non-photosynthetic organisms.

The entropy (*S_R_*) associated with radiant energy (*E_R_*) reaching the plants may be approximated as that of radiant energy diffused from the sun [14] by *S_R_* = *E_R_*/5 × 10^3^ J K^−1^. For several purposes, the ratio of energy to its associated entropy (*E*/*S*) is a measure of the quality of the energy and has a dimension of absolute temperature (K). Hence, the ratio *E_R_*/*S_R_* = 5 × 10^3^ K is a value corresponding to high-quality energy. In contrast, when *E_R_* is completely converted to heat at an ambient 300 K temperature, the new (thermal) entropy is *S_T_* = *E_R_*/3 × 10^2^ J K^−1^. The quality of the conserved energy (*E_R_*/*S_R_*) decreases to 300 K. The entropy associated with the free energy of most photosynthesized chemicals lies between a minimum of *E*/5 × 10^3^ J K^−1^ and a maximum of *E*/3 × 10^2^ J K^−1^, differing by a factor of 15.7.

Photosynthesis saves a small fraction (less than 1%) of the absorbed radiant energy as biomass supporting the following reaction [15]: 6 CO_2_ + 6 H_2_O → C_6_H_12_O_6_ + 6 O_2_
(1)
Δ*G*_0_ = 2.88 × 10^6^ J glucose mol^−1^.

Divided by the standard entropy loss in the formation of glucose, 1.16 × 10^3^ J K^−1^ glucose mol^−1^ [4], the free energy gain in photosynthesis as bonds in the glucose molecules has a quality of 2.88 × 10^6^/1.16 × 10^3^~2.5 × 10^3^ K K, which is lower than that of radiant energy but far above that of heat energy. Although differing among the high variety of metabolites and macromolecules, the 2.5 × 10^3^ K ratio may be a reference quality of the energy stored in cell molecules. However, in contrast to radiation energy, Δ*G*^0^ and Δ*H*^0^ are widely used in bioenergetic bibliographies and, in comparison with photo-physics bibliographies, are calculated by:Δ*G*^0^/Δ*S*^0^ = (Δ*H*^0^/Δ*S*^0^) − T.

Most of the radiation energy absorbed by the leaf is dissipated as heat for water transpiration [16]. Even at best, approximately 77% of the energy radiation absorbed by the photosynthetic machinery is dissipated as heat. Thus, assuming a minimum 50 photons needed to photosynthesize one molecule of glucose, photosynthesis converts 10^7^ J radiation energy (2 × 10^3^ J K^−1^ entropy) to recover 2.88 × 10^6^ J as free energy of one glucose mol endowed with 1.16 × 10^3^ J K^−1^ entropy, approximately half of that in the used radiation. Considering the entropy associated with heat, the photosynthesis results in an increase in entropy:((10^7^ − 2.88 × 10^6^)/300) − 2 × 10^3^ − 1.16 × 10^3^ ~ 2 × 10^4^ J K^−1^ glucose mol^−1^
which is only indicative because no correction of concentrations has been applied to the free energy and entropy standards and, in most cases, the photosynthesis of glucose requires more than 50 photons per molecule [17]. Notwithstanding, the 2 × 10^4^ J K^−1^ glucose mol^−1^ supplies a reference value of the minimum production of entropy associated with photosynthesis. Sato [18] calculated lower, but in the same order (1.15 × 10^4^ J K^−1^ glucose mol^−1^), entropy production considering the use of 48 photons and slightly lower entropy of radiation. Light excess over the capacity of the photosynthetic machinery increases the production of entropy through nonphotochemical quenching (NPQ) by monomeric dispersed photosystem II (PSII) and light harvesting (LHCP) complexes. These seem to assemble under low light intensities to multimeric macro-complexes that, through hiding involving pigments, decrease energy dissipation (entropy production) through zeaxanthin in NPQ [19].

Most of the entropy produced in photosynthesis takes place at the photophysical stages in the light-harvesting complexes and the photosystems, due to the absorption of photons to charge separation. The latter occurs by the transfer of one electron excited in one chlorophyll dimer to one monomeric chlorophyll and then to pheophytin [20]. As the entropy content of most metabolites is in the same range as glucose on a carbon atom gram basis, energetic considerations show that next electron transfers and pumping of protons in thylakoid, as well as conventional enzyme-catalyzed reactions in chloroplast and cytosol, account for a minor fraction of the 2 × 10^4^ J K^−1^ glucose mol^−1^ of entropy produced in photosynthesis. Then, when compared with the first “Élan Vital” [18] of photosynthesis, the changes in entropy associated with metabolic reactions are very low, falling in the range of +10 to −30 J K^−1^ mole^−1^, as deduced from Δ*G* and Δ*H* data [4] of representative reactions. 

## 3. Structural and Metabolism Entropy

The entropy change associated with the folding of the polypeptide chain to form the three-dimensional structure of the protein has been estimated experimentally and theoretically. The reported values vary within a one order of magnitude range [21,22,23,24,25]. Typical values are −1.25×10^3^ J K^−1^ mol^−1^ conformational entropy for mean globular proteins. However, the decrease in entropy due to protein folding is accompanied by a similar or higher increase in the translation entropy of the solvent water molecules [26], which leaves a negligible global effect of protein folding on the entropy balance of the living cell. In other systems, Jia et al. [27] investigated changes in entropy in the transition of lamellar to grana stacked thylakoid and concluded that it is driven by an increase in entropy. Therefore, evidence suggests that assemblages of proteins and lipids in supra-macromolecular complexes are entropy driven and that they account for more entropy to the cellular medium than their unfolded or dispersed components.

When the DNA double strand melts, entropy increases by approximately 50 J K^−1^ (mol bp)^−1^ [28]. However, the low number of DNA molecules and the entropic increase due to the small molecules crowding the DNA molecules make the possible entropy decrease associated with the double-stranded structure of DNA or, in general, the aggregation of components of the genetic machinery, negligible [29]. Thus, similarly to protein folding and lipid assemblage, secondary DNA and RNA folding does not significantly contribute to a low entropy distinction to life. 

The compartmentation of metabolites within the different cell organelles and between cells and extracellular medium implies a decrease in entropy, which was evaluated in the range of 1.0 J K^−1^ L^−1^ below the hypothetical homogeneous solute distribution [6]. Compared with the standard entropy of formation of the biomass in situ, −1.6 × 10^3^ J K^−1^ L^−1^, the compartmentation of metabolites, although essential for life [30], barely decreases the entropy of living matter by one-thousandth of the negative standard entropy of formation of their molecular components. As there is no evidence that the folding and assemblage of macromolecules contribute in a higher proportion than compartmentation to the low entropy of living matter, the question is still whether other cell structures significantly contribute to the supposed low entropy of organisms [1,31].

Adult organisms absorb nutrients and metabolize them to products that are excreted. Despite the turnover of its components, the mass and entropy of the organism open system remain constant. However, the metabolism inside produces entropy, mainly as heat and, in a lower amount, as chemicals that have more entropy than the nutrients. An adult human body may produce 10^7^ J day^−1^ as heat, carrying 4.8 × 10^2^ J K^−1^ L^−1^ day^−1^ entropy. In other words, in one hour, the human body exports, with heat, approximately 20-fold the small entropy deficit associated with the subcellular compartmentation that is key for life. Otherwise, the heat produced would duplicate the body temperature in one day (from 36.5 to 73°). Obviously, heat must be quickly exported (dissipated) to avoid membrane disassembly, protein denaturation, and cell death. Entropy export is a consequence of the high entropy of heat; it has no connection to the low entropy of compartmentation. In contrast to mass and energy, entropy is not conserved, nor can it be transferred. Except for the entropy of radiation, it is a state function that depends on the distribution of energy within molecules, and it can only increase over time. The frequently used expressions “imported” and “exported” entropies are not truly correct because entropy as such is not transferable. The organism exchanges heat energy and mass that have associated entropy. In this way, entropy stays constant in the organism and increases in the environment. The entropy that increases in the environment is not extracted from the structures of the organisms that remain unchanged. The heat and the final molecules produced in the metabolism are exported, carrying their high entropy content.

Obviously, metabolic reactions are not in equilibrium and then these non-equilibriums have associated low entropies intrinsic to organisms, which have been poorly investigated. The lower entropy associated with the non-equilibrium is cancelled at equilibrium and should be equal to the increase in entropy produced when the equilibrium is reached. Thus, the metabolism intrinsic entropy of one organism is a measure of how far the whole metabolism of the organism is from equilibrium. At equilibrium, almost all intermediaries of the whole metabolism have been converted to products, the Δ*G* of the reaction equals 0, and entropy reaches the maximum value. The metabolism intrinsic entropy (*S_i_*) of a living tissue may be estimated as the negative value of the entropy gained when the equilibrium of the whole metabolism is reached, approximately: Δ*S* = Δ*G*/*T*.

The main question is what cell components must be considered for the evaluation of the Δ*G* from a live body to the metabolic equilibrium of a dead body. One possibility is to consider only the intermediaries subjected to rapid metabolic turn-over. However, the wide range of metabolite turn-over and the variety of intermediaries and concentrations allow the gross approximation of Δ*G*. Thus, stored starch and triglycerides are not viable to calculate Δ*S_i_* in an organ such as the liver. To compare with the contribution of compartmentation to low entropy associated with life, consider the entropy associated with the continuous metabolization of glucose to CO_2_:C_6_H_12_O_6_ + 6 O_2_ → 6 CO_2_ + 6 H_2_O   Δ*G*_0_ = −2.88 × 10^6^ J glucose mol^−1^⋯ (2)

Discarding the effect of concentrations of substrates and products on Δ*G*, the entropy decrease associated with the non-equilibrium of metabolizing 5 mM glucose at 308 K would be as follows:2.88 × 10^6^ × 5 × 10^−3^/308 = 47 J K^−1^ tissue L^−1^.

This is approximately 50-fold higher than the decrease in entropy associated with compartmentation but only 2.5-fold higher than the rate production of entropy by the human body per hour, which is calculated as approximately 20 J K^−1^ L^−1^ h^−1^ (Figure 1).

The calculations are grossly approximated (possibly within one order size), but they emphasize the relevance of the intrinsic metabolism entropy and the need for its further accurate calculation, as defined here, because it is probably the major contributor of the low entropy of organisms. 

When the whole cell metabolism approaches equilibrium, intrinsic metabolism entropy increases and becomes 0 at death. In the sequential reactions of a metabolic pathway (e.g., glycolysis), enzyme inhibition, such as glyceraldehyde-3-phosphate dehydrogenase by iodoacetamide [32], leads the precedent reactions to equilibrium, but holds off equilibrium in the following reactions; thus keeping intrinsic metabolism entropy transitorily low until cell death. Metabolism is one characteristic feature of living tissue that is linked to a relatively low entropy and, thus, it contributes variably to decreasing the total entropy of life.

## 4. Production of Entropy

Metabolism produces heat that is transferred outside of chemicals with their associated entropy, which is produced at a rate, P, [33]:P = d*S*/d*t* = ∑*v_i_A_i_*/*T*(3)
where summation ∑ extends to all reaction rates, *v_i_*, and affinities, *A_i_*, as given by: *A_i_* = ∑*n_i_μ_Ri_*–∑*n_i_μ_Pi_*, where *μ_i_*s are the chemical potentials of substrates, R*_R_*, and products, P*_P_*, of the metabolic reactions *i*: ∑ R*_Ri_* → ∑ P*_Pi_*

The rate of production of entropy differs widely among organisms and physiological states, increasing, according to Equation (3), with the metabolic rate and the affinity of the global metabolic reaction. The higher the affinity and rate, the farther from equilibrium the reaction is. Therefore, departure from equilibrium has opposite effects on the rate of production of entropy that increases, and in the content of entropy of the organism that decreases due to the negative contribution of the intrinsic metabolism entropy. When approaching equilibrium, rates, *v_i_*s, become linearly dependent on chemical potentials, *μ_i_*s, and the rate of production of entropy, P, can only decrease [33,34].

In response to variable environments, the open thermodynamic systems of organisms change the separation from equilibrium of specific metabolic pathways and, sometimes, of the whole organism metabolism; thus affecting its rate of entropy production and its entropy content within ranges compatible with life. Life-compatible ranges vary among organisms and decide to be alive through evolutionary selection.

## 5. Evolution and Entropy

Relations between biological evolution and entropy have often been investigated. Organized structures and functions are characteristic of life, and evidence suggests that they became increasingly complex during the evolution of organisms. Evidently, the higher organization and complexity are supposed to imply lower entropy. This would imply the paradox that during the evolution of living beings, the entropy of biomass decreases, in contrast to the second principle of thermodynamics. The paradox appears from the ambiguous, when not arbitrary, identification of organization and complexity with low entropy and high information, and of entropy with disorder [35,36,37]. Then, information-based models of organisms propose that evolution is associated with the increased organism diversity and entropy of ecosystems [38,39,40,41] and higher entropy production [42,43] and, often, that entropy production would be maximized in fully evolved enzymes [44]. In contrast to barely quantifiable qualities (such as order and complexity) in molecular and cell biology, others, such as information and entropy, that are quantifiable and statically based, must be analyzed and distinguished [45]. In light of this, several alternative models support the trend of lower rates of the production of entropy by organisms during evolution [34,46,47,48,49,50]. 

Interpreted statistically, entropy is a measure of the uncertainty of the distribution of energy according to the equation of Boltzmann and Shannon:*S* = −*k* × ∑ *p_i_* × ln *p_i_*
where *k* is the Boltzmann constant (1.381 × 10^−23^ J K^−1^) and *p_i_* is the probability of one distribution, *i*, of the total energy among different molecules, electron excitations, bond vibration energy, etc. The statistical interpretation of entropy as a characteristic distribution of energy is relevant in biological issues, such as the understanding of the entropy content of different biomolecules.

One similar formulation is used in information theory, and the so-called informational entropy measures the uncertainty of one statement or information of a system. Informational entropy analysis is often used in biology, but its meaning should be distinguished from thermodynamic entropy in molecular biology and evolution. The genetic information in DNA supplies straightforward examples for informational entropy concepts. The characteristic nucleotide sequence of the four bases (adenine A, guanine G, cytosine C, and thymine T) in the DNA of one organism is the same in all cells of the organism, and results in the combination of random mutations and functional selection during biological evolution. There is no preference for a specific sequence; 4^n^ different DNA sequences are equally possible and the actual DNA sequence has only a 4^−n^ probability (p), where “n” (the number of bases in the DNA sequence) ranges from one million in bacteria to billions in many animals and plants. Hence, according to the Shannon formula [51,52,53], the evolutionary events resulting from an unspecified base sequence to the sequences of modern organisms result in a gain of information (a decrease in entropy, negative Δ*S*_DNA_) in “bit”:Δ*S*_DNA_ = ∑ *p* log_2_ *p* = ∑4^−n^ log_2_ 4^−n^ = 4^n^ × 4^−n^ log_2_ 4^−n^ = −2n; or Δ*S*_DNA_ = −2n bit 

Then, 2n bit is the informational entropy loss (information gain) associated with the choice of the specific DNA sequence of n bases.

The accumulation of mutations in the cells of one multicellular organism or in individuals of one species increases the informational entropy of the organism or the species, respectively. In the last case, natural selection will cut most mutants, again evolutionarily decreasing the informational entropy or genetic diversity of the species, also named populational entropy [39,54]. 

Informational and thermodynamic entropies are not equal, although the two are statistically based [55]. Then, as statistics alone cannot justify thermodynamic entropy determinants in evolutionary biology, less informational entropy could explain biological evolution without reference to the physiological and physico-chemical properties of the organisms. In the case of DNA, the key question of the choice of one specific sequence is not only a statistical issue, but also a molecular biology issue with its physico-chemical and thermodynamic determinants.

The evolutionary transition of procaryotic to eucaryotic decreased the thermodynamic entropy of the new organisms [6] due to the added compartmentation of metabolites, which look like a type of organized system. As shown in Section 3 and Section 4, the compartmentation-dependent decrease in entropy is small when compared with the variable decrease in metabolism entropy, but it is measurable and sufficiently stable to be considered characteristic of living beings and their evolution. The paradox that the highly compartmentalized eucaryotic organisms were selected, although they had lower entropy than their predecessors, seems unescapable. A closer look at the evolution of the rate of production of entropy could resolve the paradox.

As Equation (3) shows, in the stationary state of the open system of organisms, the rate of production of entropy is P = d*S*/d*t* = ∑*v_i_A_i_*/*T*, where *v_i_* is the rate of consumption of the nutrient substrate by the organism *i* and *A_i_* the affinity of the global reaction in the organism *i*. If proliferating organisms compete for the same nutrient, this becomes limiting, and the reactions approach equilibrium, when *v_i_*s depends linearly on affinities *A_i_*s. Under these conditions, the rate of entropy production per unit of time in the complete system of competing organisms cannot increase; it can only decrease to a minimum [33] per mole of constant supplied nutrient. As highlighted in [34], the decrease in entropy in a system of competing organisms may be conducted through the rapid proliferation of organisms that produce entropy at the lowest rate and the progressive disappearance of organisms that produce entropy at a high rate; the opposite is not possible (Figure 2). This is the evolutionary choice of organisms producing entropy at the lowest rate, at least under limiting conditions, which supplies a thermodynamic foundation for the evolution of organisms by natural selection. Therefore, the evolutionary trend to lower rates of entropy production in an organism system implies that the entropy of organisms per mass unit of consumed substrate should decrease in the evolution.

To decrease the rate of production of entropy, enzymes have evolutionarily acquired metabolic controls, avoiding futile cycles such as that of phospho-fructose-kinase (PFK) and fructose 1,6-bisphosphatese (FbisPasa) (Figure 3A) in glycolysis and gluconeogenesis (https://microbiochem.weebly.com/gluconeogenesis.html, accessed on 13 March 2022), respectively, which by avoiding their simultaneous activity, decrease the production of entropy through continuous waste of ATP.

Sometimes, decreases in the rate of production of entropy was reached in evolution through the elimination of metabolic routes that became unnecessary. Thus, the synthesis of ascorbic acid was lost. The loss of ascorbic acid synthesis of 30–40 Mya (Figure 3B) in the line of anthropoid primates [56] was due to the accumulation of inactivating mutations in the gene encoding L-gulono-γ-lactone oxidase, which catalyzes the synthesis of L-ascorbic acid. It is highly likely that the intense herbivore feeding of these ancestral primates supplied enough L-ascorbic acid to make its synthesis dispensable; thus saving entropy production in anthropoids. Many other examples of the conservation, elimination, and recovery of specific genes during evolution can be easily explained [56,57,58,59] by their consequences of decreasing the rate of production of entropy, as predicted by the Prigogine theorem [33,34].

The entropy produced by proliferating organisms is usually simplified as that of their main metabolism. For example, in the respiratory consumption of glucose by yeast, the only reaction considered is:C_6_H_12_O_6_ + 6 O_2_ → 6 CO_2_ + 6 H_2_O(4)
where low-entropy substrates (glucose, C_6_H_12_O_6_, and oxygen, O_2_) are converted to high-entropy products (CO_2_ and H_2_O); thus increasing the production of entropy due to the heat produced in the reaction. However, due to the proliferation of organisms, new individuals must be added to the right side of reaction (4) as products with entropy. Then, the true rate of entropy production should be lower to match the low entropy (when compared with CO_2_ + H_2_O) of the new individuals [1,34]. Consequently, the theory predicts that the selection of, with other factors equal, organisms with a lower content of entropy (possibly with increased structure–function organization) would be preferable to those with high entropy. 

## 6. Cancer and Entropy

Cancer has been a recurrent theme to confront entropy models of development, mostly of informational entropy. The development of higher plants and animals from the single cell zygote to fully differentiated adult organisms implies the growth and construction of new anatomically and functionally organized structures that, supposedly, have less entropy than the original zygote on a mass unit basis. Cancer cells deviate from the ordered development by rapid cell multiplication without specialized differentiation. Anatomically, cancer tissue appears to be disorganized and, therefore, has high entropy. The assignment of high entropy to cancer tissue seems sound and, with the correct definition of information, models of cancer growth have been linked to the expected increases in informational entropy and decreases in information [60,61] and to higher [62] or lower [63] rates of thermodynamics entropy production. However, the relation of cancer anatomy and growth to thermodynamic entropy is not as clear, and it has difficulties, as previously mentioned, in evaluating the entropy decrease associated with protein folding or the supra-macromolecular structures of the cell.

Another bioenergetic approach to cancer focuses on the Warburg effect. Cancer cells fermentatively metabolize glucose to lactate at a high rate when compared with non-cancer cells, that mainly consume glucose via respiration to produce CO_2_ [64,65,66]. Typically, cancer cells metabolize glucose fermentation by 95% [67], while healthy liver or kidney cells ferment only 15% glucose [68]. The preferent fermentative metabolism in cancer is known as the Warburg effect. The fermentative metabolism, which only yields two moles of ATP per glucose mole, is, at first glance, surprising for rapidly growing cancer cells when compared with non-cancer cells, which presumably have a lower demand of energy and consume glucose by respiration yielding 36 moles of ATP per glucose mole. The association of fermentative metabolism with rapid cell proliferation was highlighted [67] in cancer cells and microorganisms. To put this into perspective, with current ATP yields, it must be noted that respiration still accounts for approximately 50% of the ATP synthesized in typical cancer cells. Therefore, although cancer cells could metabolize 4–20-fold more glucose by fermentation than by respiration, they reach, by respiration, between 80 and 48% ATP [65].

The molecular basis of the physiological switch of glucose, from mainly respiratory to mainly fermentative, as well as the transformation from normal to cancer cells, has been investigated intensely [64,65,66,69,70], fitting them within the genetic and metabolic reprograming of cancer, but the advantages conferred to cancer cells by the fermentative metabolism have not been explained in depth.

The Warburg effect has been considered as an early and distinctive sign of cancer cells [70] linked to the stem cell model and the genetic instability of cancer cells. 

The enhanced metabolism of pyruvic acid due to lactate dehydrogenase (LDH) to produce lactic acid (Figure 4) is essential for the Warburg effect. It decreases the metabolism of pyruvic acid to acetyl-CoA by pyruvic dehydrogenase (PD) and its further respiratory consumption. Consequently, the concentration of lactic acid increases in cancer tissue, and the metabolic inhibition of LDH decreases the rate of tumor progression [69,70]. 

The association between the Warburg effect and cancer is thermodynamically intelligible [71] in a model where the total rate of entropy production tends to a minimum [72] in agreement with the Prigogine principle.

Within a thermodynamics approach, in [73], it was suggested that a lower rate of entropy production of the fermentative metabolism of glucose could supply a selective advantage for the proliferation of cancer cells. At the usual temperature (37° = 310 K), pH 7.4, concentrations (5 × 10^−3^ M glucose; 2.9 × 10^−3^ M lactate) and CO_2_ (380 ppm = 38 Pa) in human tissues, the entropy produced, per glucose mol metabolized, is lower in fermentation than in respiration:respiration: Glucose + 6 O_2_ → 6 CO_2_ + 6 H_2_O   Δ*S* = 403.9 J K^−1^ glucose mol^−1^
fermentation: Glucose → 2 Lactate + 2 H^+^   Δ*S* = 359.4 J K^−1^ glucose mol^−1^
which is a consequence of the lower entropy of lactate than of CO_2_.

Thus, like competing organisms for limited nutrients in evolutionary biology, the fermentative metabolism of cancer cells allows them increased proliferation over non-transformed cells to conduct the trend of the tissue mass to a lower rate of entropy production. Lactate is a low-entropy side product that benefits the cells producing it when competing for limiting glucose.

## 7. Concluding Remarks

Since the book of Schrödinger [1], the relation between life and entropy has been a matter of discussion and speculation barely shadowed by the impressive advances in molecular biology. Organisms supposedly have low entropy, although their activity produces entropy for the environment. Today, progress in the understanding of the nature and production of entropy parallels molecular biology insights, providing a scientific background for the two questions raised by the proposal of Schrödinger on organismal entropy: (a) How can we quantify low entropy? (b) What molecular and structural features account for low entropy? In addition, the distinction between informational and thermodynamic entropies and the rate of production of entropy was revealed to be key to understanding the dynamic of life.

Remembering that entropy is an energy associated variable, photosynthesis, the first stage leading to life, downgrades sun energy to glucose-associated energy whose entropy may be estimated to be approximately 1.6 × 10^4^ J K^−1^ carbon kg^−1^ or 1.6 × 10^3^ J K^−1^ per liter (L) of an alive or dead body. Energy conversion in photosynthesis conforms to the thermodynamic second principle and barely reaches 2–3% efficiency, while it increases entropy by approximately 2.8 × 10^5^ J K^−1^ carbon kg^−1^, which is mainly produced in the photo-physical stages of light energy conversion.

By taking the 1.6 × 10^3^ J K^−1^ L^−1^ of life mass as a reference, the assumed low thermodynamic entropy distinguishing alive from dead biomass seems to be associated with decreases in the range of 1 J K^−1^ L^−1^, due to the compartmentation of metabolites, and 40–50 J K^−1^ L^−1^ estimated for the departure of metabolic reactions from equilibrium. The decreases are slight but demand, like those of the structural and informational designs of metabolites and macromolecules, further precise quantifications to define the limits between health and pathology. The two are temperature sensible, which compels the continuous export of heat with its associated entropy. Intense experimental and theoretical investigations suggest that there is no other living feature that contributes significantly to the low entropy associated with life.

Recent investigations on the rate of production provide entropy with added relevance in the molecular biology of evolution and development. The central question is whether organism metabolism tends to maximize or minimize the rate of production of entropy. The two possibilities have been theoretically and experimentally proven for specific biological systems and non-living models. The trend to minimize the rate of production of entropy is based on the theorem of Prigogine, which seems to be applicable for organisms or cells competing for one nutrient and confirms the choice of those producing entropy at the lowest rate and the elimination of those that produce entropy at a high rate. The minimization model could resolve the long-standing question of the physical bases of the evolution by natural selection and supply the thermodynamic background to understand the rapid proliferation of cancer cells.

## Figures and Tables

**Figure 1 ijms-23-04098-f001:**
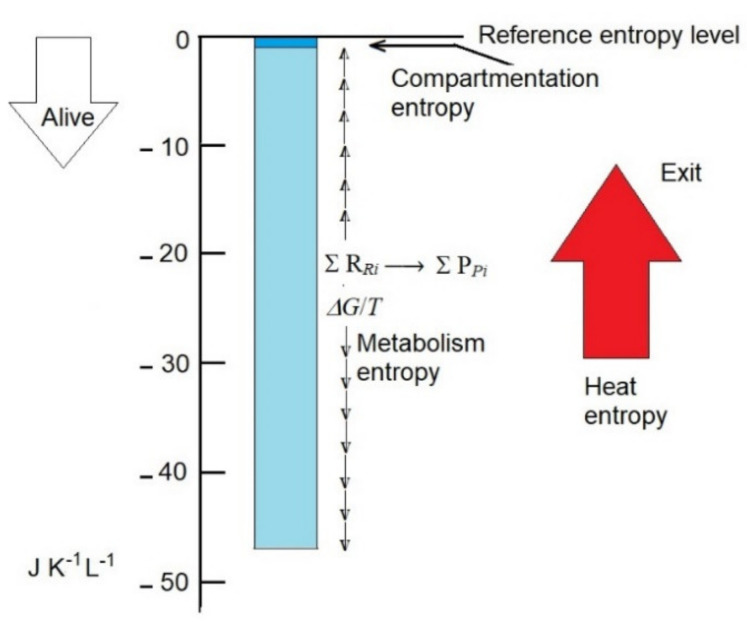
Compartmentation entropy and metabolism entropy are plausibly the main contributors that, by decreasing the entropy of non-living biomass (reference entropy level), convert it to live biomass. Entropy values in the scale may be representative of the human body. Compartmentation entropy is essentially constant, and metabolism entropy (due to the departure from reaction equilibrium) is variable. Exportation of heat entropy (red arrow) prevents the collapse of compartmentation and metabolism entropies to 0.

**Figure 2 ijms-23-04098-f002:**
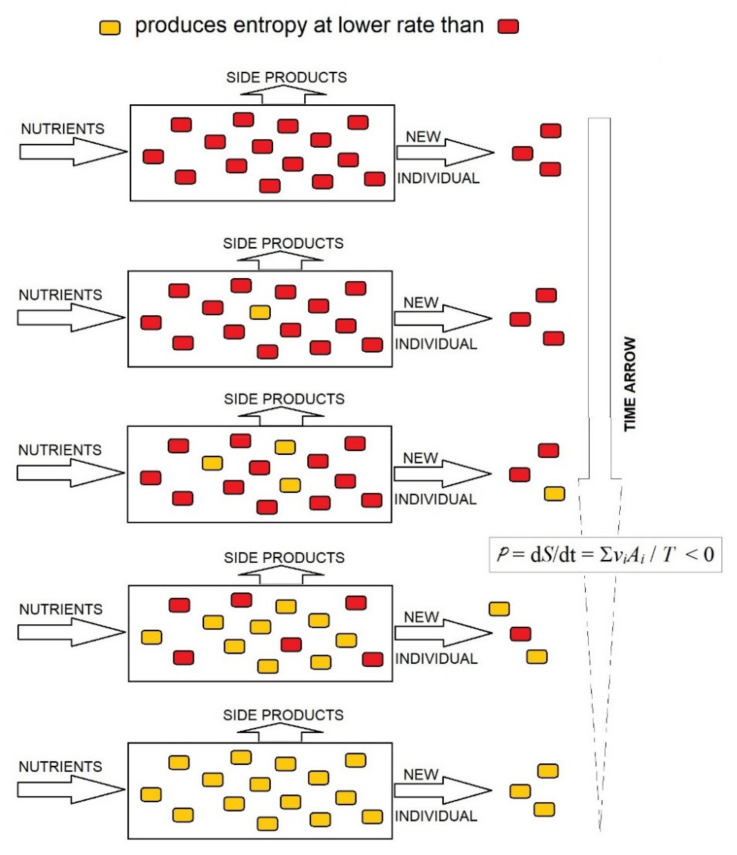
Decrease in the rate of production of entropy in an open system saturated by organisms. Under a limited supply of nutrients, the system is close to equilibrium and can only evolve to decrease the rate of production of entropy, which is achieved by the progressive substitution of organisms that produce entropy at a high rate (red
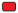
) with organisms producing entropy at a low rate (yellow
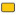
).

**Figure 3 ijms-23-04098-f003:**
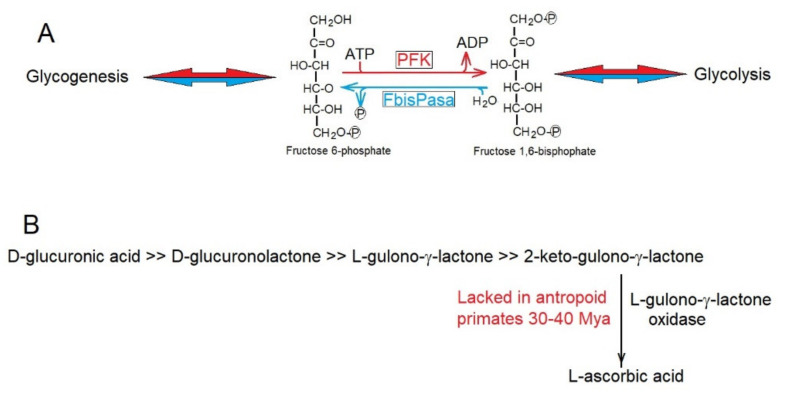
(**A**) Phospho-fructose-kinase (PFK) and fructose 1,6-bisphosphatase (FbisPase) would provide a futile metabolic cycle in glycolysis and glycogenesis consuming ATP and producing entropy. Strict metabolic controls ensure that they do not function simultaneously. (**B**) Biosynthetic pathway for the synthesis of L-ascorbic acid. Anthropoid primates lack the enzyme L-gulono-γ-lactone oxidase.

**Figure 4 ijms-23-04098-f004:**
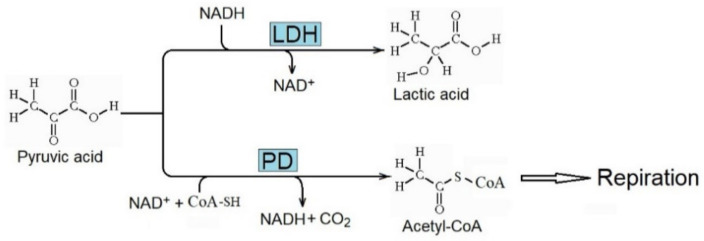
Conversion of pyruvic acid to lactic acid and acetyl-CoA catalyzed by lactate dehydrogenase (LDH) and pyruvic dehydrogenase (PD), respectively. Enhanced activity of LDH is critical for the Warburg effect and the production of lactic acid in cancer.

## Data Availability

Not applicable.

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
