# Peer review of "Entropy Perspectives of Molecular and Evolutionary Biology"

_ijms, 2022, doi:10.3390/ijms23084098_

Round 1
Reviewer 1 Report
In the manuscript titled “Entropy Perspectives of Molecular and Evolutionary Biology”, the author has provided their thoughts on the relationship between the entropy of an organism and their evolution. Through these thoughts, they then attempt to hypothesize how their evolution is regulated by the entropic cost of their metabolic processes.
The discussion provided to in the author’s review is sound and coherent to follow. The topic is very relevant and I recommend that this manuscript should be considered for publication. However, I have some concerns regarding the formulation of several sentences in the manuscript. I will outline my concerns in detail below:
-
The manuscript should be benefited by another read through. There are several grammatical and syntactical errors in the text. E.g. “… physic(al) significance which determines …” on page 1, “These seem to assemble .. “ on page 3, etc. to name a few.
-
The sentence “Then, supposedly, living features additional to that of chemical biomass should account for minor contributions to low entropy of organisms, however they are decisive for life.” on page 2 is hard to follow.
-
The manuscript will significantly be benefited by adding more references and citations, particularly when a new though or a keyword (like negentropy) is introduced. This will also help distinguish author’s thoughts from the community’s.
-
On page 2 author mention ratio of energy (E/S). This term is not entirely clear. Particularly, I am not sure which energy are we talking about here. In the example provided, is this the radiant energy diffused from the sun, or the energy incident on a plant (which does photosynthesis) from the sun. It might be useful to provide a clearer definition and example.
-
The figure 2 provided is very intriguing, but the connection provided between the rate of entropy production and the rate by which resources are consumed is not very clear. There are several unaccounted factors in the equation like stoichiometry to name a few. Also, how does this relate to the entropy of the organism is not explained carefully. This part of the manuscript will benefit if further explanation along with citations is provided.
Author Response
Thank you very much for your encouraging criticism. I think that the new version addresses your concerns and improves clarity and scientific relevance of the manuscript. After correction through the MDPI English Editing Services (https://www.mdpi.com/authors/english), my changes/additions are red labelled in the new version.
- and 2. According your specific suggestions, I have revised and corrected sentences in pages 1, 2 and 3. After service editing, I carefully read and correct the whole manuscript and red labelled the new added text.
- Accordingly, three new citations are included in the new version. I would wish to include more pertinent publications, however, at the present, they require extensive text additions that would distort the whole manuscript. The new references and the changes in the citing numbers are red labelled.
- As now red labelled (page 2), E/S ratio firstly refers to radiant energy diffused from the Sun. For quality comparation beyond photosynthesis, the manuscript extents the concept to other energy forms as detailed in the English edited paragraph.
- I hope that the connection between the rate of entropy production and the rate by which resources are consumed (to accomplish the theorem of the minimum entropy rate production) is now detailed in the new red labelled words in the last paragraph of page 8. Changes and additions in the last paragraph of page 9 further details the relation between entropy production rate and resource consumption. The stoichiometry factor is involved because coefficients in Equation (4) should be adjusted for the formation of one new complete organism.
Reviewer 2 Report
Brief summary
The manuscript by Sabater summarize some molecular and structural features of biomolecules that contribute to the low entropy associated to life. The author discusses the poorly understood role of entropy in biological processes. Although there is much still to be understood about how low entropy of a living organism is maintained, the review has collected some useful and relevant data, which highlights the significance of entropy in molecular and evolutionary biology. I really enjoyed reading the review paper. The implications of thermodynamics in natural selection and cancer cell survival are fascinating and further research on these areas will be useful for biologists to study evolution. However, some issues need to be considered before publication.
Comments
- The line numbers are missing, which would be extremely helpful for review process.
- The review paper needs extensive paraphrasing of the statements. In many instances, it is not clear what the author is trying to say.
- A thorough reading by a native English speaker would clarify the manuscript.
- The author needs to explain in a better way why entropy contribution to physiology is important, instead of just saying it would be beneficial to use entropy approaches to investigate development, cancer, and biological evolution.
- Author could add how free energy or enthalpy usage is not sufficient to determine the course of biological processes, and why entropy would be better parameter to look at.
- Please either add something or remove …, from the second paragraph (page 2).
- Please clarify how -1.6 x 103 J K-1L-1 is calculated in second paragraph (page 2)
- It is Glucose not glucosa in equation 1 in page 3, and equation 2 in page 5
- How loss of function of paralogs in higher organisms can be explained with evolutionarily decreasing informational entropy during natural selection? Some losses of functions are beneficial to the organisms.
- Compartmentalization in eukaryotic cells is more or less similar to each other. How can entropy explain that humans are superior than for example dogs and are evolutionarily advanced organism?
- Can we conclude that low metabolism can reduce the entropy of an organism, and is beneficial during evolution? Is there anything known about total metabolic activity of organisms that are at different levels in an evolution tree?
Needs a major revision.
Author Response
Thank you very much for your stimulating words and criticism that, I hope, is satisfactorily addresses in the revised version of the manuscript. After correction through the MDPI English Editing Services (https://www.mdpi.com/authors/english), my changes/additions are red labelled in the new version.
- Sorry! I also prefer line numbers but introduce them now will not help.
- and 3. I carefully read and correct the English and content of the whole manuscript before and after correction through the MDPI English Editing Services. I hope that now the manuscript message become clear.
- and 5. I try to make clearer the usage relevance of free energy, enthalpy, and entropy in biology through new point (red labelled) texts in the last paragraphs of page 1, and page 9. In addition to the improved whole text editing, I hope that now the specific utility of entropy to understand developmental and evolutionary questions appears promising enough.
- Please, see the new text. I hope that the new edited text and the red labelled additions make clear the second paragraph of page 2.
- Please, see details in the parenthesis (red labelled) inserted immediately in the second paragraph of page 2.
- “glucose” has now been corrected to “glucose” in Equations 1 and 2.
- Of course, some losses of function are beneficial under specific environments. The example described in the manuscript of the loss of ascorbic acid synthesis is one among many that may be documented at molecular and genetic detail and where informational entropy approaches would provide new perspectives.
- Compartmentation advantage was one marker of low entropy content that was selected early in biological evolution. Specialized tissue structures and functions are tempting generalization of the same process but, in my opinion, these possibilities should be treated carefully. Caution must be paid to avoid subjective concepts. For example, humans are not necessarily superior, at least thermodynamically, in all aspects and environments. Here, questions of brain energetics, waste activities and, of course, informational entropy would be addressed.
- Low metabolism does not necessarily reduce the entropy of an organism and is beneficial during evolution. As discussed in the manuscript, the reference is the rate of entropy production per mole of nutrient necessarily consumed to form a new individual of the specie. Total metabolic activity is usually negatively correlated with size of the specie. However, the size has increased or decreased in a phylogenetic line, probably dependent on the physical and chemical changing environment and, also probably, the rate of entropy production has a key role selecting organism size under specific environment.
Round 2
Reviewer 1 Report
The addendums made to the manuscript are appropriate and the author has added clarity to the manuscript. I recommend the manuscript to be published in its current form.
Reviewer 2 Report
The author has significantly edited and improved the manuscript. The author has addressed all the questions appropriately.
I enjoyed reading this review paper.
The paper should be good for publication now.
This manuscript is a resubmission of an earlier submission. The following is a list of the peer review reports and author responses from that submission.